# Mitochondrial Dysfunction in Cardiac Diseases and Therapeutic Strategies

**DOI:** 10.3390/biomedicines11051500

**Published:** 2023-05-22

**Authors:** Yafei Huang, Bingying Zhou

**Affiliations:** State Key Laboratory of Cardiovascular Disease, Fuwai Hospital, National Center for Cardiovascular Diseases, Chinese Academy of Medical Science and Peking Union Medical College, 167 North Lishi Road, Xicheng District, Beijing 100037, China; huangyafei@fuwai.com

**Keywords:** mitochondria, cardiomyocytes, cardiovascular diseases, mitochondrial dysfunction, therapeutic strategies targeting mitochondria

## Abstract

Mitochondria are the main site of intracellular synthesis of ATP, which provides energy for various physiological activities of the cell. Cardiomyocytes have a high density of mitochondria and mitochondrial damage is present in a variety of cardiovascular diseases. In this paper, we describe mitochondrial damage in mitochondrial cardiomyopathy, congenital heart disease, coronary heart disease, myocardial ischemia–reperfusion injury, heart failure, and drug-induced cardiotoxicity, in the context of the key roles of mitochondria in cardiac development and homeostasis. Finally, we discuss the main current therapeutic strategies aimed at alleviating mitochondrial impairment-related cardiac dysfunction, including pharmacological strategies, gene therapy, mitochondrial replacement therapy, and mitochondrial transplantation. It is hoped that this will provide new ideas for the treatment of cardiovascular diseases.

## 1. Introduction

The heart is the first organ to form and function during mammalian embryonic development, and its development is essential for the maturation of the cardiovascular system and the formation of other organs. Mitochondria produce ATP to drive cardiomyocyte contraction, thereby providing energy for the heart to pump blood. In addition, mitochondria are key regulators of the cardiomyocyte response to various stimuli such as hypoxia, oxidative stress, and hyperglycemia, and are involved in a wide range of biological functions (e.g., regulation of calcium and reactive oxygen species (ROS) signaling) [1,2,3]. Mitochondrial dysfunction has been associated with abnormal electron transport chain (ETC) activity, reduced ATP production, abnormal transfer of metabolic substrates, overproduction of ROS, increased mitochondrial DNA (mtDNA) damage, cristae disruption, and metabolic defects [4]. Mitochondrial damage is prevalent in cardiovascular diseases (CVDs), and timely correction of mitochondrial dysfunction and improvement of energy production deficits early in the disease becomes critical in the treatment of CVDs.

## 2. Mitochondrial Function in Mammalian Cardiac Development

The morphological formation and developmental remodeling of the heart is a precise and coordinated process. The maturation of cardiomyocytes prepares mammals for powerful, efficient, and sustained pumping throughout the mammalian life cycle, which is essential to meet the functional and metabolic needs of a growing heart. In the process of mammalian heart development and maturation, the maturation and metabolism of mitochondria play a key role [5].

### 2.1. Embryonic Stage

At the embryonic stage, anaerobic glycolysis is the preferred route of energy production. Embryonic day 9.5 (E9.5) mouse hearts contain relatively few and immature mitochondria, characterized by rare and disordered cristae, along with a low mitochondrial membrane potential (MMP), high levels of ROS, and open mitochondrial permeability transition pores (mPTP). In contrast, in E13.5 mouse hearts, mitochondria significantly increase in mass and evolve into large round organelles rich in cristae [6,7]. In this process, the shutdown of mPTP alters mitochondrial oxidative metabolism and redox signaling, leading to reduced levels of ROS that drive the maturation of mitochondrial structure and function, thereby inducing cardiomyocyte differentiation [8,9,10]. In addition, mitochondria also promote embryonic heart development by increasing the content of mtDNA and regulating Ca^2+^ signaling [11].

During cardiac development, several regulatory factors of mitochondrial biogenesis and dynamics play important roles. The transcriptional co-activator peroxisome proliferator-activated receptor gamma coactivator 1 alpha (PGC-1α) is a master regulator of mitochondrial biogenesis and cardiac energy metabolism. PGC-1α plays an important role in heart development. PGC-1α^−/−^ mice exhibited abnormal mitochondrial content, increased markers of cardiac dysfunction, compromised ATP production, and diminished cardiac inotropic responses [12,13,14]. Another regulator of mitochondrial biogenesis, *Tfam*, was also shown to be pivotal to intact cardiac function. Targeted inactivation of *Tfam* in embryonic cardiomyocytes was lethal, which was the direct result of elevated ROS products, DNA damage, and severely suppressed cardiomyocyte proliferation [15]. Mitofusin 2 (MFN2) is a mitochondrial dynamics-related protein that is primarily involved in the mitochondrial fusion process. Ablation of *Mfn2* in embryonic mouse hearts arrested cardiac development [16]. Removal of *Mfn2* in mouse embryonic stem cells (ESCs) impaired ESC differentiation into cardiomyocytes [16]. MFN2 was found to be enriched at mitochondrial-associated membranes (MAMs), which form the segments of the endoplasmic reticulum (ER) that are tethered to mitochondria, regulating the morphology of ER and directly linking the two through trans-organelle interactions to enhance mitochondrial Ca^2+^ uptake efficiency [17,18]. Ablation or silencing of *Mfn2* in mouse embryonic fibroblasts disrupted the shape of the ER and its tethering to mitochondria, thereby reducing mitochondrial Ca^2+^ uptake and transport [18].

In addition to some of the above regulators of mitochondrial quality control, proteins involved in mitochondrial protein turnover also play an important role in embryonic heart development. For example, the mitochondrial matrix AAA+ Lon protease (LONP1), a multifunctional enzyme involved in various aspects of mitochondrial protein turnover, was shown to be indispensable for normal cardiac development. In LONP1-deficient mouse cardiomyocytes, levels of proteins associated with the ETC were significantly reduced [19]. Its deletion resulted in embryonic lethality and was marked by mitochondrial swelling, loss of cristae, as well as abnormal accumulation of protein aggregates. The Snf2-related CREBBP activator protein (SCRAP) complex is an ATP-dependent chromatin remodeling complex that regulates the incorporation of histone variant H2A.Z into chromatin. The SCRAP complex was found to be essential to normal embryonic mitochondrial maturation. Znhit1 is one of the core subunits of the SRCAP complex and has been demonstrated to play an essential role in supporting the function of the SRCAP complex [20]. Disruption of the SRCAP complex by deletion of *Znhit1* in embryonic cardiomyocytes led to impaired heart development at E13.5. The left ventricular myocardium was thinned and dilated, leading to heart failure (HF) and prenatal or perinatal lethality. At the subcellular level, *Znhit1* deletion impairs the integrity of the SRCAP complex, resulting in mitochondrial swelling and crest damage. At the protein level, the expression of the mitochondrial respiratory chain subunit components Ndufb8, SDHB, Uqcrc2, complex II, and ATP5A of Znhit1-KO mouse hearts was significantly inhibited from E11.5 to E18.5, and ATP production in mutant hearts was also significantly reduced. However, at the transcription level, no significant reduction in mRNA for the above respiratory chain subunits was observed. These results suggested that the SCRAP complex plays a role in maintaining the mitochondrial protein turnover necessary for normal cardiac development [21].

### 2.2. Postnatal Stage 

After birth, mammalian cardiomyocytes undergo a number of maturation changes associated with increased cardiac function and output, including hypertrophic growth [22], cell cycle withdrawal [23], myosin isoform conversion [24], and mitochondrial maturation [25,26]. Piquereau et al. revealed the link between the maturation of energy pathways and cellular structure within cardiac myocytes during the development of the mouse heart from birth to adulthood [27]. Mitochondria between myofibrils in 3-day-old mouse cardiomyocytes were irregularly dispersed and misaligned, and there was a large cytoplasmic space between myofibrils and myofilaments. The ultrastructure of cardiomyocytes from 7-day-old mice showed markedly reduced cytoplasmic spaces and increased mitochondrial clusters arranged along longitudinal myofilaments. The cardiomyocytes of 21-day-old mice were full of myofibrils, and mitochondria were arranged vertically, which was similar to the appearance of adult cells. Cardiomyocytes from 63-day-old mice exhibited a regular overall ultrastructure, with myofilaments and mitochondria arranged in parallel along the longitudinal axis [27]. This constellation allows mitochondria to approach myofibrils and sarcomere networks, expand their contact surfaces, and promote Ca^2+^ transport to sustain energy supply. Mitochondrial morphology and functional maturation during postnatal development of the heart are regulated by the estrogen-related receptor (ERR) [28], Parkin-mediated autophagy [29], PGC-1α [6], etc. Detailed adaptations of mitochondria in the growing heart are reviewed in [5,30]. 

### 2.3. Mitochondrial Function in the Adult Heart

The adult heart is one of the most active metabolic organs in the body, producing roughly 30 kg of ATP per day. The approximately 6000 mitochondria in adult mammalian cardiomyocytes account for 30–40% of the entire cell volume. Cardiac mitochondria are involved in a wide range of biological functions, including the regulation of Ca^2+^ and ROS signaling [2,3,31,32,33,34].

The mitochondrion is a double membrane organelle. The outer mitochondrial membrane (OMM) is involved in the synthesis of phospholipids and precedes the initial breakdown of substances that will undergo complete oxidation in the mitochondrial matrix. The inner mitochondrial membrane (IMM) folds into cristae towards the inner lumen and contains more proteins than the OMM, and thus hosts more complex biochemical reactions. The main components of the IMM are the enzymes involved in electron transport and ATP synthesis, forming the ETC. Under normal physiological conditions, oxidative phosphorylation (OXPHOS) produces more than 95% of cellular energy in the form of ATP. Supporting the operation of OXPHOS are five enzyme complexes of the ETC, mitochondrial complexes I–V [35]. The mPTP is known to be a non-selective pore located in the IMM. As Ca^2+^ is taken in and released by the mitochondria, the mPTP switches back and forth between open and closed states [36]. ROS or Ca^2+^ overload triggers mPTP opening, causing mitochondria to swell and cells to die [33].

The mitochondrial matrix is the inner space in the mitochondria that is encapsulated by the IMM and contains numerous proteins such as enzymes involved in biochemical reactions such as the tricarboxylic acid cycle, fatty acid oxidation, and amino acid degradation. The mitochondrial matrix also contains the mitochondrion’s own DNA (mtDNA), RNA, and ribosomes. Human mtDNA is a 16,569 base pair-long closed-loop DNA molecule. The mtDNA contains 37 genes that encode 13 polypeptides, 22 tRNAs, and 2 rRNAs of the ETC to regulate OXPHOS and protein assembly [37]. The 22 species of mitochondrial tRNA (mt-tRNA) are required for the translation of essential subunits of the ETC. Its mutations, post-transcriptional modifications, and the metabolism of related enzymes are important for normal mitochondrial protein translation and homeostasis [38]. Mutations in tRNA-processing enzymes, human mitochondrial RNase P and Z, which are responsible for removing the 5’-extensions and 3’-tails of tRNAs, have been implicated in infantile hypertrophic cardiomyopathy (HCM) and HSD10 disease, characterized by progressive neurodegeneration and cardiomyopathy, respectively. For example, short-chain dehydrogenase/reductase 5C1 (SDR5C1/HSD17B10), a structural component of RNase P, was found to carry missense mutations that cause HSD10 disease. In vitro biochemical analysis of *SDR5C1* mutants demonstrated reduced endonucleolytic activity and altered interaction with TRMT10C, the methyltransferase subunit of the RNase P complex [39]. Mutations in another RNase P subunit, PRORP, also caused mt-tRNA processing deficits, and pleiotropic abnormalities, including but not restricted to, fetal tachycardia [40]. However, the direct mechanistic link between these mutations and the pathogenesis of related cardiac anomalies is as yet unclear. 

Mitochondrial quality control (MQC) is an endogenous protective mechanism that is essential for the maintenance of mitochondrial homeostasis and integrity, which are primarily the concerted actions of mitochondrial biosynthesis, mitochondrial dynamics, and mitochondrial autophagy [41]. Impairments of MQC mechanisms can cause mitochondrial dysfunction and may lead to CVDs [42,43,44]. Mitochondrial biosynthesis is a dynamic process by which new mitochondria are formed in cells to maintain and restore the mitochondrial structure, number, and function in the face of increased energy demand. Impaired mitochondrial biosynthesis is closely associated with CVDs [45]. Mitochondrial dynamics refers to cycles of fusion and fission that are critical to maintaining mitochondrial function during stress. Dynamin-related protein 1 (DRP1) is a major pro-fission protein whose activity is tightly regulated, and which removes damaged mitochondria through mitophagy, ensuring tight control of the complex processes of cardiac cell and organ dynamics [46,47]. Mitochondrial fusion in adult cardiomyocytes is necessary for the maintenance of normal mitochondrial morphology and is essential for normal cardiac respiratory and contractile function [48]. Mitofusin 1 (MFN1) and MFN2 are GTPases that act on the OMM to promote fusion. Genetic depletion of *MFN1/2* in the heart leads to mitochondrial fragmentation and fatal dilated cardiomyopathy (DCM) [49]. Loss of function mutation of *Mfn2* (M376A), however, while causing mitochondrial shortening, depolarization, and perinuclear aggregation, known as “mitochondrial clumping”, did not lead to embryonic mouse lethality, which may be due to functional redundancy between the mitofusins [50]. Mouse embryos lacking *Drp1* display mitochondrial aggregation in cardiac fibroblasts, which die after E12.5 [51]. However, overexpression of *Drp1* alone induced mitochondrial fragmentation and cardiomyocytes overexpressing *Mfn2* alone resulted in enlarged mitochondria, but they still exhibited normal mitochondrial respiration without cardiac pathology or lethality [29,52]. In addition, Song and colleagues found that a heart-specific triple deletion of *Mfn1*/*Mfn2*/*Drp1* led to unique pathological cardiac hypertrophy in mice, accompanied by defects in mitochondrial autophagy, protein imbalance, and decreased mitochondrial biosynthesis [52]. The most apparent abnormality in these triple deficient cardiomyocytes was the perinuclear accumulation of mitochondria [52]. These lines of evidence suggested that kinetic imbalance caused by overexpression of mitochondrial fission and fusion-related genes appears to have no effect on the heart, but imbalance caused by gene deletion of fission or fusion proteins alone is embryonically lethal. Mitochondrial adynamism, however, while not as lethal as an imbalance caused by deficiency of fusion or fission alone, also caused HF and death in adult animals in the long run [52].

Mitochondrial autophagy is the engulfment of damaged or dysfunctional mitochondria by autophagosomes, mainly including the PINK1–Parkin pathway and receptor-dependent mitophagy mediated by Bnip3, Nix, or Fundc1 [53]. Most CVDs are associated with either activation or inhibition of autophagy, including HF, ischemic heart disease, cardiac hypertrophy, cardiomyopathy, and ischemia–reperfusion (IR) injury [54,55,56,57].

### 2.4. Mitochondrial Adaptations with Age

During cardiac aging, systolic dysfunction, left ventricular hypertrophy, and increased cardiac fibrosis, accompanied by mitochondrial accumulation of protein aggregates, contribute to cardiac dysfunction [58,59,60,61]. The size of mitochondria increases in senescent cardiomyocytes, the internal ultrastructure is disordered, the matrix appears translucent, the number of crests decreases significantly, and they begin to lose their strict parallel position. During normal aging, mitophagy levels in mammalian tissues are significantly reduced [62]. This might account for the known accumulation of damaged mitochondria and provides a mechanism for the observed defects in mitochondrial function and increased oxidative stress in senescent tissues and organs [63,64]. In addition, mtDNA is surrounded by oxygen radicals, and its mutation rate is ten times higher than that of nDNA. With age, mtDNA mutations accumulate progressively, copy number decreases significantly, and the abundance of the mitochondrion’s own RNA (mtRNA) and the rate of mitochondrial protein synthesis decrease [65,66]. In addition, impaired mtDNA replication and excision can lead to double-strand breaks, due to the unique replication mechanism as well as the lack of DNA damage repair system in mtDNA, thus causing massive base deletions [67]. In a single-cell sequencing study in the elderly, more than 90% of mtDNA mutations were found to be localized in protein-coding genes and tRNA gene regions, and the vast majority (>75%) of protein-coding gene mutations were nonsynonymous mutations. While direct evidence of aging-induced alterations in mtDNA-encoded subunits is scarce, we postulate that the mutations and copy number alterations that occur during aging will lead to aberrancies in ETC subunit expression [68]. The accumulation of massive mtDNA deletions leads to an increase in ROS and free radicals, while the function of associated free radical scavenging enzymes is reduced, and thus the balance between pro-oxidative stimulation and antioxidant defense is lost [69,70]. Increases in ROS production, in turn, exacerbate OXPHOS impairment, leading to a loss of metabolic fitness and, ultimately, myocardial abnormalities during aging [61,71]. At the same time, protein aggregates and lipofuscin appear in cardiomyocytes. Protein oxidation and ubiquitination increase, while autophagic activity decreases, leading to the accumulation of harmful cellular waste, thereby producing a positive feedback loop, gradually aggravating oxidative stress and mitochondrial dysfunction [60,72,73,74,75,76]. Many of these alterations in aging cardiomyocytes are also present in CVDs, suggesting common molecular bases between cardiac aging and CVDs [77]. 

In addition, a gradual decline in mitochondrial activity leads to heterogeneity and dysregulation of key metabolites that also extensively affect the aging process [78,79]. Recently, Wu et al. found that the mitochondrial unfolded protein response (mitoUPR) transcription factor ATFS-1 promotes the lifespan of mitochondrial mutants by activating stress response pathways in *C. elegans* [80]. In prematurely aged mice, the NADH-AMPK-p53-dependent pathway drives mitochondrial dysfunction-associated senescence (MiDAS) and causes a pronounced secretory signature and mitotic arrest that can be rescued with pyruvate [81]. In addition, a peptide encoded by a short open reading frame in mtDNA, mitochondrial-derived peptide (MDP), plays a key role as a cytoprotective factor in the cellular stress response [82]. Age-related diseases are associated with reduced plasma levels of MDP, such as coronary artery disease [83], Alzheimer’s disease [84], and type 2 diabetes (T2D) [85]. Therefore, a better understanding of mitochondrial dysfunction and aging and age-related diseases might provide a basis for reducing morbidity in later life in the future.

## 3. Mitochondrial Dysfunction in Cardiovascular Diseases

A growing number of studies have shown that mitochondrial dysfunction is closely associated with CVDs. Below we summarize the specific mitochondrial abnormalities in a variety of CVDs, including mitochondrial cardiomyopathy, congenital heart disease, coronary heart disease, myocardial IR injury, and drug-induced cardiotoxicity.

### 3.1. Mitochondrial Cardiomyopathy

Mitochondrial cardiomyopathy (MCM) is a genetic disease caused by mutations in mtDNA, which leads to disturbances in mitochondrial protein homeostasis and defects in ETC enzymes, thereby interfering with mitochondrial energy production to directly affect cardiac function [86,87]. 

MCM is frequently caused by mutations in mt-tRNAs [88]. Failure of mt-tRNA metabolism usually results in a decrease in the activity of the ETC complexes. Recently, some mt-tRNA mutations associated with myocardial infarction have been summarized and described. For example, the m.8306T>C mutation in the *tRNA^Lys^* gene disrupts the thymine-adenine (T-A) bond in the D-arm of tRNA, the m.3243A>G mutation that causes inefficient aminoacylation of *tRNA^Leu^* (UUR), and the m.4317A>G mutation that causes inefficient hetero-nucleation of *tRNA^Ile^* [88].

Mutations have also been identified in many other genes, including GTP binding protein 3 gene (*GTPBP3*) [38], ubiquinone oxidoreductase subunit A7 gene (*NDUFA7*) [89], ubiquinone oxidoreductase subunit B11 gene (*NDUFB11*) [90], mitochondrial small ribosomal protein 14 gene (*MRPS14*) [91], elaC ribonuclease Z 2 gene (*ELAC2*) [92], tafazzin gene (*TAZ*) [93], and DnaJ heat shock protein family member C19 gene (*DNAJC19*) [94].

#### 3.1.1. Hypertrophic Cardiomyopathy

Mutations in genes associated with the ETC complex result in impaired ATP production and inadequate energy supply to myocardial tissue. As a result, degenerative changes and compensatory hypertrophic hyperplasia occur in the myocardium, leading to HCM [95]. In the transverse aortic constriction (TAC)-induced HCM mouse model, the expression of genes involved in pathways such as calcium signaling, oxidative stress, and energy metabolism is altered. Gene Ontology analysis revealed that the expression of mitochondrial ribosomes and phosphorylated genes was closely related to the degree of cellular hypertrophy, and that transcription and translation of nuclear genes involved in OXPHOS were rapidly induced in this process [96]. Mitochondrial aberrancies in HCM often involve genetic and functional defects of ETC complex I, particularly in the *Ndufa7* gene. The expression of *Ndufa7* was significantly decreased in mice subjected to TAC, a model of pressure overload-induced hypertrophy [89]. *Ndufa7* depletion led to marked impairment of ventricular function in developing zebrafish embryos, with concomitant upregulation of hypertrophy markers *Nppa* and *Nppb* [89]. Other complex I and IV defects arise indirectly as a result of disturbed mtDNA replication and protein translation. Gene mutation in thymidine kinase 2 (*TK2*), a gene required for proper mtDNA replication, was identified as the underlying cause in a pediatric HCM patient [97]. ETC complex activities of mitochondria isolated from this patient showed a profound reduction in the activities of complex I and IV, with decreased ratio to citrate synthase. Likewise, in another pediatric HCM patient harboring a mutation in the alanyl-tRNA synthetase 2 (*AARS2*) gene, severe reductions in both the activities and amounts of complexes I and IV, were observed. *MRPS14* mutations were found to disrupt mitochondrial function by perturbing translational elongation or mitochondrial mRNA recruitment. Interestingly, biochemical analyses revealed that this mutation specifically induced complex IV deficiency that was not ameliorated by other supportive treatments [91].

Mt-tRNAs also play a critical role in the pathogenesis of early onset HCM. *ELAC2* encodes the long isoform of the human tRNA processing nuclease RNase Z, which is responsible for the removal of the 3’ tail during tRNA biogenesis and is therefore involved in tRNA maturation [98,99]. Mutations in *ELAC2* have been linked to HCM. *ELAC2* mutations were identified in five cases of infantile HCM. The abundance of unprocessed tRNA precursors was substantially enhanced in mutant fibroblasts derived from these individuals, which was rescued by the exogenous supply of wild-type *ELAC2*. At the molecular level, impaired protein synthesis of mitochondrial proteins, including complex I deficiency, was identified as a potential mechanistic basis of such disease [100]. In another cohort of 13 infants, Saoura and co-workers identified 16 novel *ELAC2* variants linked to ETC deficiency, HCM, and lactic acidosis [92]. Accumulation of unprocessed tRNA was evident in patient-derived primary fibroblasts, but further mechanistic evaluation of its functional consequences, or how it related to cardiomyopathy, was not provided. An improved understanding was achieved in the *Drosophila* model system, where cardiomyocytes displayed increased multinucleation and ploidy in *ELAC2*-mutant hearts, which might explain the increased cell size present in HCM [101]. Additionally, mutant hearts also demonstrated increased extracellular matrix (ECM) deposition, which may also contribute to HCM [92]. However, studies in human samples have not yet inspected nucleation and ploidy of cardiomyocytes, or deposition of ECM, aspects that would be worthwhile to look into in future efforts. GTPBP3 is a conserved mt-tRNA modifying enzyme that catalyzes 5-taurinomethyluridine (τm5U) biosynthesis at the 34th wobble position of mt-tRNAs. Genetic mutations in *GTPBP3* alter the folding and structural stability of mt-tRNAs, rendering tRNAs ineffective in recognizing codons during protein translation, generating protein homeostatic stress, and changing the activity of the ETC complexes. In *Gtpbp* knockout zebrafish, the embryonic heart developed abnormally, with progressive hypertrophy of cardiomyocytes and disruption of cardiac myogenic fibers, eventually manifesting as HCM [102]. Respiratory complex I, II, and IV all exhibited significant declines in their protein levels, whereas activities were reduced for complexes I–IV. Thus, mt-tRNA dysregulation is associated with cardiac disease, and precise control of mt-tRNA is an essential regulatory mechanism for coordinating mRNA translation and protein expression in cardiac homeostasis [103].

#### 3.1.2. Dilated Cardiomyopathy

Defects in the mitochondrial ETC and metabolite circulation (tricarboxylic acid cycle and lipid oxidation) in cardiomyocytes may also lead to ventricular systolic dysfunction, resulting in DCM [104]. 

A number of mutations in mtDNA have been identified in DCM. In an mt-tRNA mutation screen of 318 DCM cases, seven potentially pathogenic mutations were identified: MT-TL1 3302A>G, MT-TI 4295A>G, MT-TM 4435A>G, MT-TA 5655T>C, MT-TH 12201T>C, MT-TE14692A>G, and MT-TT 15927G>A [105]. Compared to control patients, patients carrying these mutations had significantly lower mtDNA copy numbers, reduced ATP production, and increased ROS production, which directly correlated with changes in energy reserves, oxidative stress, and MMP. Of these, MT-TL1 3302A>G caused severe defects in ETC complex I and reduced complex IV activity. The 4295A>G and 4435A>G mutations affected the highly conserved adenosine at position 5 in the MT-TI and MT-TM anticodon stems, respectively, impairing the aminoacylation capacity of tRNAs. However, the exact mechanism by which these mutations lead to DCM is largely unknown. One explanation is that structural and functional abnormalities caused by mt-tRNA mutations impair mitochondrial protein synthesis and respiration, leading to oxidative stress and decoupling of oxidative pathways from ATP synthesis, resulting in DCM.

There is also growing evidence of private mtDNA mutations in the pathogenesis of DCM. In a large cohort study of 601 patients with DCM, 19 heterogeneous mtDNA mutations (i.e., 9 tRNA, 5 rRNA, and 5 missense mutations, cf. Table 1 in [106]) were identified from 85 patients with abnormally increased number and size of myocardial mitochondria as well as displaying aberrant ultrastructure. Using a trans-mitochondrial cybrid system, Govindaraj and co-workers evaluated the functional consequences of two private mutations, m.8812A>G and m.10320G>A, and revealed significant decreases in MMP, elevated ROS, suppressed oxygen consumption, as well as cell death [107]. Due to the large proportion of idiopathic DCM cases, it is conceivable that the application of genome-wide sequencing techniques, such as whole exome sequencing (WES), will uncover increasingly more private mutations as a powerful tool in identifying rare and private mutations, that are currently undetectable via panel testing [108]. The challenge remains, however, in discriminating the disease-causing mutations from the vast number of background polymorphisms.

Dysregulation of mitochondria function-related proteins encoded by nuclear genes also plays a prominent role in DCM. Transcriptome analysis of left ventricular tissue in a cardiomyocyte-specific mammalian sterile 20-like kinase 1 (Mst1)-overexpressing mouse model of DCM revealed significant downregulation of the 13 mtDNA-encoded mRNAs. Meanwhile, transcriptional activators encoded by *Tfam, Tfab1, Tfab2,* and *Polrmt* also showed consistent downregulation. Likewise, nDNA-encoded genes involved in mt-rRNA processing were also downregulated in DCM mice. In the same vein, the majority of nDNA-encoded mitochondrial aminoacyl-tRNA-synthetases, which govern tRNA function, were decreased in DCM. In addition to the regulation of mtDNA, the expression of genes that regulate mitochondrial protein import machinery was similarly suppressed. These findings indicate that perturbations in nuclear genes that control mitochondrial homeostasis are potential triggers of the DCM phenotype [109]. More definite evidence comes from a pediatric patient with DCM with ataxia syndrome (DCMA) who harbored a homozygous pathogenic mitochondrial *DNAJC19* variant. *DNAJC19* encodes mitochondrial import inner membrane translocase subunit TIM14, and its expression was significantly reduced in cardiomyocytes. Mitochondrial abnormalities included the increased presence of scattered electron-dense inclusions, the molecular identities of which remain unknown [94]. A typical example of a DCM phenotype associated with a nuclear gene defect is Barth syndrome (BTHS), an X-linked recessive disorder caused by mutations in the gene encoding the mitochondrial transacylase (i.e., *TAZ*), which leads to cardiolipin remodeling and thus mitochondrial dysfunction [93].

### 3.2. Congenital Heart Disease

Congenital heart disease (CHD) is an abnormality in the morphology, structure, and function of the heart due to abnormal cardiovascular development during the embryonic period.

In search of CHD-related mtDNA mutations, Abaci and colleagues performed next-generation sequencing on myocardial samples from 22 CHD patients, and identified 13 previously unknown mutations in *ATP6*, *CYTB*, *ND5*, *ND4,* and *ND2* genes [110]. 

Other CHD patients, while not directly harboring mutations within the mitochondrial genome, still suffer from the deleterious consequences of mitochondrial defects, which occur secondary to hemodynamic distortions. Several clinical studies have found widespread mitochondrial dysfunction through the study of tissue samples from patients undergoing congenital heart surgery [111,112]. Patients with CHD commonly suffer from the chronic consequences of a pressure-overloaded right ventricle (RV). HF eventually occurs when the RV is no longer able to compensate. In a study of 31 myocardial specimens (25 with RV hypertrophy and 6 with RV failure) without mitochondrial mutations, progressive mtDNA depletion was found as one of the initiating steps towards mitochondrial dysfunction, which marked the transition to HF [113]. Specifically, this depletion was attributed to defective mtDNA replication and was likely regulated in an RV pressure-dependent manner [113]. The left heart hypoplasia syndrome (HLHS) is a serious CHD with polygenic and genetic heterogeneity. Mitochondria in the left ventricular myocardium exhibited abnormal shape and size, along with reduced numbers of cristae, indicative of mitochondrial maturation defect [114]. However, whether mtDNA mutations are drivers of CHD is controversial. A recent study suggested that neither mtDNA mutations nor their copy number variations contribute significantly to the pathogenesis of CHD [115]. Cyanotic CHD, also called critical heart disease, is a severe type of CHD in which malformation reduces the amount of oxygen supplied to the body. In a comparative analysis of cyanotic and acyanotic CHD patients, myocardial biopsies from cyanotic CHD patients demonstrated reduced mitochondrial densities and ATP production, suggesting that the degree of mitochondrial dysfunction is associated with disease severity [111]. Even as a consequent pathology, mitochondrial impairment can serve as a potential therapeutic target in the treatment of congenital heart defects.

### 3.3. Coronary Artery Disease

The pathological basis of coronary artery disease (CAD) is coronary intimal damage, lipid deposition following impaired endothelial cell barrier function, and smooth muscle proliferation, along with the secretion of cytokines that promote a chronic inflammatory response, culminating in the formation of atheromatous plaques.

Mt-tRNA mutations had been previously identified in CAD [116,117]. In maternally inherited CAD patients, a *tRNA^Thr^* mutation (15927G>A) was identified that changed the conformation of the tRNA. This resulted in an 80% decrease in steady-state *tRNA^Thr^* abundance, mitochondrial protein synthesis defect, mitochondrial respiration deficiency, and elevated ROS. Similarly, sequencing analysis of the mitochondrial genome of another patient diagnosed with inherited CAD identified 45 mutations, including 19 silent mutations, 14 D-loop variants, 8 missense mutations that affect protein-coding genes, 2 12S rRNA variants, a 16s rRNA variant, and a homozygous 15910C>T mutation in the *mt-tRNA^Thr^* gene. Mutant cybrid lines harboring the 15910C>T mutation showed significantly reduced *tRNA^Thr^* and mitochondrial protein levels, impaired complex I and III activity, decreased ATP production, and increased ROS production [117].

Additional indirect pathogenic mechanisms of CAD that involve mitochondria have also been reported. For example, mitochondria have emerged as a player in CAD through their potential interconnections with profilin. Mutations in profilin, an actin-binding protein, result in the distorted cytoskeleton and mitochondrial morphology and impaired mitochondrial mobility, thus impairing its fusion and fission functions [118,119]. Profilin phosphorylates p53, which then translocates to the mitochondria to activate other apoptotic pathways. At the same time, the profilin-SIRT3 interaction impairs the ETC and MMP, leading to increased ROS levels [119].

Furthermore, hypertension is a major risk factor for CAD [120]. The pathogenesis of obese hypertensive populations is associated with altered mitochondrial dynamics, ETC, and ROS production. Mitochondrial genetic variants were found in a rat model of hypertension in genes encoding ETC proteins, mt-tRNA, and genes related to tricarboxylic acid (TCA) metabolism, resulting in dysfunctional mitochondrial energy production [121]. The IMM fusion protein OPA1 was significantly downregulated in a hypertensive mouse model, leading to increased mitochondrial fission and oxidative stress, increased vascular cell apoptosis, and reduced vascular smooth muscle cell proliferation, which predisposes to CAD [122].

### 3.4. Myocardial Ischemia–Reperfusion Injury

During myocardial IR, free radical production is increased due to mitochondrial OXPHOS disorders, xanthine oxidase catalysis, and neutrophil respiratory explosion. At the same time, excessive production of ROS in mitochondria destroys the mitochondrial membrane structure, resulting in the release of cytochrome c in large quantities, thereby initiating apoptosis through the mitochondrial pathway [123].

In an ischemic or hypoxic heart, the expression of many mitochondria-related enzymes and genes is increased, which causes mitochondrial dysfunction in cardiac muscle cells leading to myocardial damage [124,125,126]. In response to ischemic stress, proprotein convertase subtilisin/kexin type 9 (PCSK9) expression is strongly upregulated in the border region of infarcted mouse hearts, and autophagy ensues [124]. Ischemia-induced ROS triggers the feed-forward cycle of PCSK9 upregulation, which in turn aggravates ROS accumulation. ROS inhibitors diphenyleneiodonium (DPI) and apocynin significantly inhibited *PCSK9* expression, while *PCSK9* inhibition (Pep2-8/EGF-A or siRNA silencing) reduced ROS production, suggesting bidirectional crosstalk between ROS and PCSK9. In addition, PCSK9 and the microtubule-associated protein 1A/1B-light chain 3 (MAP1LC3, LC3B) were also found to be highly expressed in the border region in heart sections of patients who died of acute myocardial infarction, indicating the conservation of these mechanisms. Further studies found that activation of the ROS-ATM-LKB1-AMPK axis was a possible mechanism of PCSK9-induced autophagy. Mice pretreated with Pep2-8/EGF-A demonstrated a 20–30% reduction in the infarct area, improvement (approximately 25%) in systolic function, significant reduction in autophagic activity, and much thinner left ventricular wall in the area around the infarct zone than in wild-type mice undergoing left coronary artery (LCA) ligation [124]. *PCSK9* knockdown in adult rats alleviated myocardial injury induced by occlusion of the left anterior descending (LAD) branch. This protective effect was attributed to the repression of the Bnip3 autophagy pathway and inhibition of the inflammatory response [126]. 

However, under other circumstances, mitophagy is required to clear cells of defective mitochondria during IR damage. In mouse models, cardiac IR insult induced the expression of *CK2α*, the latter phosphorylated and thus inactivated the FUN14 domain containing 1 (FUNDC1), thus suppressing mitophagy in effect. Defective autophagy led to severe mitochondrial dysfunction, marked by the collapse of the mitochondrial genome, inhibition of ETC complexes, stalled mitochondrial biogenesis, cardiolipin oxidation, ROS overproduction, mPTP opening, and ultimately facilitating cardiomyocyte apoptosis [125]. Time-course transcriptome profiling in a mouse model of IR revealed elevated S100a8/a9 as an early marker of IR, while removal of this protein conferred protection from IR damage [127]. S100a8/a9 inhibited the expression of the mitochondrial *NDUF* genes by nuclear respiratory factor 1 signaling, thereby inhibiting mitochondrial complex I activity, causing mitochondrial respiratory dysfunction in cardiomyocytes [127]. 

### 3.5. Heart Failure

Heart failure is a group of clinical syndromes in which ventricular filling or ejection dysfunction is impaired due to structural or functional abnormalities of the heart and is the end stage in the development of many CVDs. Mitochondrial dysfunction is closely associated with the development of HF [128]. 

Damaged mitochondria are strong activators of inflammation. The NLRP3 inflammasome is a well-characterized pathogenic factor of HF [129]. NLRP3 senses mitochondrial dysfunction, particularly mitochondrial ROS generation. In a specific type of HF, namely HF with preserved ejection fraction (HFpEF), increased assembly of NLPR3 was observed on hyperacetylated mitochondria. Increasing β-hydroxybutyrate level attenuated both inflammasome formation and mitochondrial acetylation and rescued HFpEF phenotypes in mice [130]. Therefore, in these circumstances, mitochondrial dysfunction drives HF pathogenesis through inflammation.

Further upstream, abnormalities in mitochondria-associated proteins or enzymes have been reported as underlying causes of mitochondrial dysfunction that lead to HF [131]. A-kinase anchoring protein 1 (AKAP1) is localized to the OMM and is involved in regulating mitochondrial function by interacting with the NADH-ubiquinone oxidoreductase 75 kDa subunit (NDUFS1). In a diabetic mouse model of HF, loss-of-function and gain-of-function experiments revealed that *Akap1* deficiency inhibits ETC complex I activity by blocking the translocation of NDUFS1 from the cytoplasm to the mitochondria, leading to mitochondrial dysfunction and cardiomyocyte apoptosis [132]. The expression of protein kinase DYRK1B was significantly upregulated in both failed human myocardium and hypertrophic mouse hearts. DYRK1B binds directly to STAT3 to increase phosphorylation and intranuclear aggregation, thereby inhibiting the expression of PGC-1α, a key regulator of mitochondrial energy production, ultimately leading to impaired mitochondrial energy production and HF in cardiomyocytes [133].

### 3.6. Drug-Induced Cardiac Toxicity

Drug-induced cardiotoxicity (DIC) is a complex variety of pathophysiological damage to the cardiovascular system caused by drugs, which manifests clinically as systolic or diastolic dysfunction and cardiac arrhythmias. Mitochondrial damage is a common form of cardiotoxicity. Many drugs cause mitochondrial dysfunction through inhibition of ETC complex activity, ROS production, lipid peroxidation, and glutathione and ATP depletion, such as non-steroidal anti-inflammatory drugs (NSAIDs), beta-blocker drugs, selective cyclooxygenase-2 inhibitors, arsenic trioxide (ATO), anthracyclines, etc. [134,135,136].

Cardiotoxicity caused by antineoplastic drugs is well studied, and its manifestations are complex and varied, commonly including arrhythmias, myocardial ischemia, coronary artery disease, hypertension, and myocardial dysfunction. The main chemical categories of antineoplastic drugs that commonly cause cardiotoxicity are anthracyclines (e.g., doxorubicin, erythromycin, etc.), alkylating agents (e.g., cyclophosphamide, etc.), antimetabolites (e.g., fluorouracil), drugs that interfere with microtubule protein synthesis (e.g., paclitaxel), and targeted drugs (e.g., trastuzumab). Among them, treatment with anthracyclines is prone to both acute and chronic cardiotoxicity, which can lead to impaired left ventricular function and HF. CVDs are the leading cause of morbidity and mortality in cancer survivors treated with anthracyclines, largely limiting the clinical use of these drugs. In clinical cancer patients, anthracycline-induced cardiotoxicity is characterized by myocardial damage caused by pathophysiological changes such as protein hydrolysis, necrosis, apoptosis, and fibrosis [137,138,139]. A series of toxicity studies of doxorubicin (DOX) at both organ and cellular levels in rats and mice also confirmed mitochondrial morphological and structural abnormalities, such as mitochondrial swelling, membrane collapse, crest loss, and matrix cavitation [140]. ATO, a drug used to treat leukemia, increased intracellular calcium levels, caused alterations in the activities of transcription factor Nrf2, xanthine oxidase, aconitase, and cysteine 3, and cells exhibited increased oxidative stress, reduced innate antioxidant status, mitochondrial dysfunction, and apoptosis [141].

Non-steroidal anti-inflammatory drugs (NSAIDs, celecoxib, valdecoxib, rofecoxib, etc.), used to treat pain and inflammation are another class of drugs that may increase the risk of heart attack or stroke. NSAID-induced cardiac cytotoxicity is caused by the induction of apoptotic signaling via a ROS-mediated shift in mitochondrial permeability [142]. Celecoxib, a selective cyclooxygenase-2 inhibitor, causes a decrease in complex IV activity accompanied by MMP collapse, ROS formation, mitochondrial swelling, ATP depletion, and lipid peroxidation [143].

Long-term use of antipsychotics (AP) is a common cause of myocardial injury and even sudden cardiac death, and AP treatment has been reported in the clinical and preclinical literature, and AP treatment is associated with increased cardiometabolic abnormalities, ventricular arrhythmias, and sudden cardiac death in patients [144]. Mitochondrial impairment is associated with the pathophysiology of a variety of psychiatric disorders, and the use of AP further leads to mitochondrial dysfunction and decreased ATP production [145]. In a study measuring the effects of mitochondria on three typical Aps and seven atypical Aps on mitochondrial bioenergetic in pig brains, all tested Aps (i.e., zotepine, aripiprazole, quetiapine, risperidone, and clozapine) significantly inhibited mitochondrial complex activity except olanzapine [146]. A preclinical animal study illustrated the effects of AP on the cardiovascular system. The authors explored the mechanism by which AP increased CVD risk in risperidone-treated mouse hearts and found that risperidone altered the characteristics of the mouse heart proteome. Among them, mitochondrial respiratory complex I and proteins involved in mitochondrial function and OXPHOS pathways were expressed differently, and the oxygen consumption of the mitochondria in the heart, as well as the consumption of systemic energy, were also altered [147]. However, cardiotoxicity studies of AP drugs have mainly focused on their proarrhythmic risks, while the detailed mechanisms of their mitochondrial liabilities remain to be further explored [148].

Several cardiovascular drugs also exhibit mitochondrial toxicity [149]. For example, vasodilators (such as organic nitrates, molsidomine, etc.), can stimulate the production of ROS, thereby causing mitochondrial oxidative damage. Side effects such as nitrate tolerance and endothelial dysfunction occur during long-term treatment with vasodilators [150]. The release of nitrogen monoxide (NO) from vasodilators by catabolism in the circulation tends to induce ROS production in the presence of oxygen, causing mitochondrial oxidative damage and dysfunction of the ETC complex. The development of tolerance to long-term angiotensin treatment and endothelial dysfunction is also associated with increased ROS production, which reduces the bioavailability of the drug [149,151]. Antiarrhythmic drugs can also cause mitochondrial dysfunction. Quinidine reduced mitochondrial respiratory function in rat kidney cells evidenced by a decrease in respiratory control index (RCI) and ADP/O ratio [152]. In addition, quinidine was able to reduce electron transfer activity and inhibit mitochondrial protein synthesis in the mitochondria of the rat heart [153]. Lidocaine was able to induce structural changes in human neutrophils, reduce ATP and MMP, and induce apoptosis [154]. Amiodarone caused hepatotoxicity during administration in rats, which further induced cardiolipin peroxidation by increasing mitochondrial H_2_O_2_ synthesis. At the same time, it also inhibited the activity of complex I, leading to a decrease in hepatic ATP content [155]. In cultured rat vascular smooth muscle cells, verapamil reduced the perinuclear density of mitochondria, resulting in upregulation of autophagy and antiproliferative effects [156]. These drugs have also demonstrated certain cardiotoxicity in clinical practice, including, but not restricted to, hypotension, prolongation of the QT interval on electrocardiography (ECG), and abnormalities of the cardiac conduction system [157,158]. It is speculated that these adverse reactions may be related to mitochondrial dysfunction, but direct evidence of this will be needed in the future.

In summary, there is a strong link between many CVDs and mitochondrial dysfunction. We have summarized the mitochondrial dysfunction in CVDs and the associated mechanisms in Table 1. However, the current compilation of the relevant literature suggests that the mechanisms underlying the occurrence of mitochondrial dysfunction are relatively complex and that there is an interactive, causal relationship that requires further exploration and research. With further investigation, the relationship between CVDs and mitochondrial dysfunction will gradually become more apparent, providing an important reference for the future clinical management of CVDs.

**Table 1 biomedicines-11-01500-t001:** Mitochondrial dysfunction and related mechanisms in cardiovascular disease.

Dysfunctional Mitochondrial Component	Molecules	CVD	Ref.
ATP	DYRK1B, PGC-1α	HF	[133]
Autophagy, Anti-proliferative	Verapamil	DIC	[156]
Complex I	*Ndufa7*	HCM	[89]
Complex I	*NDUFB11*	HCM	[90]
Complex I	S100a8/a9	IR injury	[127]
Complex I	AKAP1	HF	[132]
Complex IV	*MRPS14*	HCM	[91]
Complex IV	Cyclooxygenase-2	DIC	[143]
Complex I and IV	*TK2*	HCM	[97]
Complex IV, OXPHOS	Risperidone	DIC	[147]
ETC	*ELAC2*	HCM	[92]
ETC	SRCAP complex	CHD	[21]
ETC	Zotepine, Aripiprazole, Quetiapine, Risperidone, Clozapine	DIC	[146]
ETC, ROS	Profilin, Profilin-SIRT3	CAD	[119]
ETC, Mitochondrial protein synthesis	Quinidine	DIC	[153]
MtDNA	*ATP6*, *CYTB*, *ND5*, *ND4*, and *ND2*	CHD	[110]
MtDNA	Replication defects	CHD	[113]
MtDNA	PCSK9	IR injury	[159]
Mt-tRNA	M.8306T>C	MCM	[88]
Mt-tRNA	M.3243A>G	MCM	[88]
Mt-tRNA	M.4317A>G	MCM	[88]
Mt-tRNA	*GTPBP3*	HCM	[102]
Mt-tRNA	3302A>G, 295A>G, 4435A>G, 5655T>C, 12201T>C, 14692A>G, 15927G>A	DCM	[105]
Mitochondrial morphology	*MFN1/2*	DCM	[49]
Mitochondrial morphology	DOX	DIC	[140]
Organelle	*TAZ*	DCM	[93]
Organelle	*DNAJC19*	DCM	[94]
Organelle	ATO	DIC	[141]
Organelle	Antiarrhythmic drugs	DIC	[152]
Organelle	MtDNA, MtRNA	DCM	[109]
Organelle	Mitochondrial density and ATP	Cyanotic CHD	[111]
Organelle	Defects in mitochondrial maturation	HLHS	[114]
Organelle	Mt-tRNA	DCM	[116,117]
OXPHOS	Vasodilators	DIC	[150]
OXPHOS	M.8812A>G, M.10320G>A	DCM	[107]
ETC, mPTP	CK2α	IR injury	[125]
ROS	NLRP3	HF	[130]
ROS	NSAIDs	DIC	[142]

## 4. Current Therapeutic Medications and Strategies

Current mitochondrially targeted therapeutic strategies focus on factors that contribute to mitochondrial damage, such as the ETC and mitochondrial dynamics. These factors are extensively involved in the progression of CVDs and therefore represent potential mitochondrial targets for CVDs treatment. Beginning in the late 1990s, an intensive wave of preclinical and clinical research has investigated mitochondrial dysfunction as a therapeutic target for CVDs [160]. However, to date, no molecules specifically targeting mitochondria have been available for the clinical treatment of CVDs [161]. There is hope, nonetheless, that breakthroughs in some of the newly developed techniques, such as mitochondrial replacement therapy (MRT), may be applied in the treatment of CVDs [162]. Herein, we summarize major therapeutic strategies for targeting mitochondria in CVDs (see also Table 2).

### 4.1. Mitochondrially Targeted Therapeutic Drugs

Mitochondrially targeted drugs for the treatment of CVDs largely focus on increasing the activity of mitochondrial complexes, suppressing the production of mitochondrial ROS, inhibiting mPTP opening, and balancing mitochondrial dynamics. While being developed, drug candidates have fallen short of expectations, mainly undermined by unsatisfactory target specificity and delivery. Therefore, a better understanding of the molecular mechanisms underlying mitochondrial dysfunction in CVDs is needed in the future to identify targets for intervention and improve drug specificity and reduce adverse effects [163].

#### 4.1.1. Drugs Targeting Mitochondrial Complexes

Drugs that act directly by modulating the activities of mitochondrial complex proteins remain scarce. Elamipretide formerly known as SS-31, MTP-131, or Bendavia, is a Szeto–Schiller peptide that selectively binds cardiolipin on the IMM, stabilizing the cardiolipin–cytochrome c super-complex. Its binding prevents the conversion of cytochrome c from an electron carrier to a peroxidase [164].

Numerous pre-clinical and clinical studies have evaluated the mitochondria-protective function of elamipretide [165,166,167]. Elamipretide improved cardiac function and prevented left ventricular remodeling in a rat model of myocardial infarction [165]. Elamipretide preserved the expression of many mitochondrial function-related genes, suppressed ROS production, and maintained the activities of complexes I and IV in the border zone [165]. The long-term effects of elamipretide were tested in a canine model of intracoronary microembolization-induced chronic HF [166]. It improved left ventricular function in both acute and chronic settings and restored all measures of mitochondrial function tested, including MMP, mitochondrial state 3 respiration, maximum rate of ATP synthesis, and ATP/ADP ratio to near normal levels [166]. Treatment of failing and non-failing ventricular tissue in freshly transplanted children and adults with elamipretide acutely resulted in significant improvements in mitochondrial oxygen flux, complex I and IV activity, and super-complex activity [168]. Due to its remarkable performance in preclinical models, elamipretide was further advanced into clinical trials. Early incremental dose clinical trials demonstrated that elamipretide caused favorable changes in left ventricular volumes in relation to peak plasma concentrations, supporting a temporal association and dose-effect relationship [169]. Currently, relevant phase 2/3 randomized clinical trials have been completed, such as evaluating the effectiveness of elamipretide in BTHS. The results of this trial showed that the use of elamipretide can lead to an increase in the patient’s left ventricular volume and increased cardiac stroke volume, which can lead to improved symptoms of BTHS [170].

In addition to the complex protective drugs mentioned above, mitochondrial complex I inhibitors, counterintuitively, also exhibit protective effects in CVDs. Metformin is the drug of choice for the clinical treatment of T2D, and it inhibits mitochondrial respiratory chain complex I to activate AMPK [171,172,173]. However, it has now been shown to exert cardioprotective effects beyond glycemic control. Sardu et al. reported that metformin reduced the risk of coronary heart disease by reducing coronary endothelial dysfunction [174]. In IR injury in rats, metformin activated the RISK pathway, maintained mPTP in a closed state, and reduced the size of myocardial infarction [175]. Despite these benefits, the dosage of metformin requires awareness. Long-term exposure to metformin can cause its accumulation in mitochondria, which can lead to mitochondrial dysfunction. Life-threatening lactic acidosis occurs when its plasma concentration reaches 32 mmol/L [176]. Another ETC complex I inhibitor, amobarbital (Amo), was shown to protect from myocardial injury by blocking electron transfer and reducing superoxide production and Ca^2+^ overload [177,178]. Thus, complex I inhibitors protect the heart through dampening potential ROS production resulting from electron flow, or by activating cardioprotective signaling pathways. However, due to the inhibition of the activity of mitochondrial complexes, these drugs may not be applicable to the treatment of acute myocardial infarction.

#### 4.1.2. Drugs Targeting Mitochondrial Redox State

Therapeutic approaches that use mitochondrial oxidative metabolism as a pharmacological therapeutic target have emerged as a very promising treatment for improving myocardial injury. OXPHOS modulators are the most widely used class of mitochondrial-targeted therapies, which include OP2113, Idebenone, Mito-TEMPOL, etc.

The active ingredient of OP2113, ATT, is a specific inhibitor of ROS production from ETC complexes. It blocks mitochondrial ROS/H_2_O_2_ production without impairing electron transfer [179]. This drug demonstrated cardioprotective properties in the acutely injured myocardial setting. In mitochondria isolated from rat hearts, OP2113 reduced ROS and H_2_O_2_ that were induced by high concentrations of succinate [180]. In in vitro IR-stimulated human skeletal muscle myoblasts, OP2113 treatment induced an increase in steady-state levels of ATP measured after reperfusion. In an in vivo rat model of IR injury, OP2113 treatment significantly reduced the size of myocardial infarcts. This drug, therefore, holds promise for the treatment of IR injury, but further in vitro studies in human cardiomyocytes, mammalian model studies, and clinical trial studies are needed [180].

Idebenone is a short-chain quinone lipophilic compound that acts as a potent antioxidant to protect mitochondria from oxidative stress. Idebenone is effective in controlling cardiac hypertrophy in patients with Friedreich’s ataxia (FRDA), a disease that has no chance of spontaneous reversal. A clinical study using it to treat FRDA patients for six months revealed a significant reduction in left ventricular weight in about half of the patients, with no serious side effects [181,182]. Idebenone is currently clinically approved, but its mechanism of action remains controversial. It has been suggested that in the treatment of myocardial IR injury, idebenone protects mitochondria by bypassing the dysfunctional ETC complex I and directly stimulating the downstream electron transport system to increase ATP synthesis, without apparent ROS scavenging properties [183]. However, it has recently been shown that idebenone regulates ROS through the ROS-AMPK-mTOR axis to modulate ROS-dependent autophagy and inhibit iron concentration for cardioprotective effects [184]. While idebenone has demonstrated great potential in the treatment of myocardial hypertrophy, myocardial infarction, myocardial infarction, and other related CVDs, further research into its diverse mechanisms of action and more CVD-related clinical studies are still needed.

Carvedilol is a β blocker with intrinsic antioxidant activity that has been described to protect the heart’s mitochondria from oxidative damage [185]. In a 2018 prospective study on the prevention of anthracycline cardiotoxicity with carvedilol, carvedilol was found to reduce troponin levels and improve cardiac systolic function in treated patients with lower rates of cardiovascular comorbidity and risk factors for cardiotoxicity [186]. Carvedilol has been used in the treatment of a small number of children with ectopic tachycardia and QT-prolonged arrhythmias, who displayed clinical relief and good tolerability [187]. Carvedilol is well tolerated and has few adverse effects, but its use in clinical practice, especially in children with CVDs, requires further studies on dosing and long-term prognosis.

Mito-TEMPOL is an antioxidant targeting mitochondria, formed by the combination of the superoxide dismutase mimetic TEMPOL and triphenylphosphine (TPP), and is now widely used as an antioxidant both in vitro and in vivo [188]. The lipophilic TPP cation enables Mito-TEMPOL to rapidly cross biological membranes, including the OMM and IMM. TEMPOL is a nitrogen–oxygen radical containing stable nitrogen oxide, which functions to scavenge oxygen radicals [189]. In cellular models, Mito-TEMPOL can be rapidly converted to Mito-TEMPOL-H, both forms of which inhibit lipid peroxidation. In addition, the in vivo reduction of TEMPOL to hydroxylamine mediates a variety of antioxidant effects. Hydroxylamine was more effective than Mito-TEMPOL in preventing lipid peroxidation and reducing oxidative damage to mtDNA [190]. By scavenging ROS, Mito-TEMPOL inhibited nicotine-induced opening of mPTP in the rat heart, suppressed cardiac hypertrophy and cardiac fibrosis, and prevented nicotine-induced cardiac remodeling and dysfunction. Its protective effect was compared against an established cardioprotectant, dexrazoxane, in a syngeneic rat model where the breast tumor cell line SST-2 was implanted into immune-competent spontaneously hypertensive rats (SHRs) treated with DOX. Mito-TEMPOL reduced the number of lesions compared to DOX only but failed to confer protection to the extent of that of dexrazoxane [191]. Aside from Mito-TEMPOL, TPP has also been conjugated with other antioxidants such as ubiquinone, tocopherol, nitrones, and plastoquinone for the treatment of targeted mitochondrial disease, and relevant studies have been conducted in Parkinson’s disease, chronic hepatitis C, and dry eye syndrome but not yet in CVDs [192]. Therefore, their effect on CVDs or DIC awaits further testing.

Propofol is an anesthetic that has antioxidant properties and was shown to improve mitochondrial function. One study found that propofol can maintain mitochondrial homeostasis by transcriptionally activating a mitochondrial protein LRPPRC, protecting cardiomyocytes from hypoxia-induced damage [193]. Diazoxide, a drug designed for hyperinsulinism was reported to negatively regulate cardiac hypertrophy by opening mitochondrial ATP-sensitive potassium channels to reduce ROS production and Ca^2+^-induced swelling, thereby avoiding mitochondrial oxidative damage [194]. It also showed promise as a cardioprotectant during cardiac surgery possibly by preventing mitochondria dysfunction [195].

#### 4.1.3. Drugs Targeting Mitochondrial Permeability Transition Pore

Cardiovascular disorders such as IR injury are associated with mPTP opening. There is ample evidence in experimental models that pharmacological inhibition of mPTP opening reduces infarct size and inhibits cellular apoptosis [196].

Cyclosporine A (CsA) is a well-known inhibitor of mPTP opening, achieved through its interaction with a critical regulator of mPTP, i.e., cyclophilin D (CypD). In a clinical trial of 58 patients with acute ST-segment elevation myocardial infarction, CsA limited infarct size during myocardial infarction and attenuated lethal myocardial injury during reperfusion [197]. In a multicenter clinical trial of IR, intravenous CsA did not achieve better frontal clinical outcomes than the placebo group, which could be attributed to improper timing of drug administration and coexisting conditions [197]. Therefore, accurately identifying the potential caveats of CsA administration is essential for promoting better clinical outcomes.

Because the high molecular weight of CsA (1202.61 g/mol) limited its bioavailability, small-molecule cyclophilin inhibitors of CypD were sought after. C31, a small molecule CypD inhibitor, ameliorated CaCl_2_-induced mitochondrial swelling and improved calcium retention. It inhibited mPTP opening and cell death in both H9c2 cardiomyoblasts and isolated mouse cardiomyocytes, demonstrating potential as a promising cytoprotective agent for the treatment of IR injury, but methods to improve its myocardial distribution are necessary to further exploit the clinical potential of this compound [198].

TRO40303 is a new cardioprotective drug that is specific to the mitochondrial translocator protein 18 kDa (TSPO) and delays the opening of mPTP. Unlike CsA, it has no effect on the calcium retention capacity of cardiomyocyte mitochondria [199]. TRO40303 reduced oxidative stress in hypoxic–reoxygenated cardiomyocytes in vitro and reduced the size of infarcts in preclinical animal models [200]. TRO40303 has already entered phase I clinical trials, with no incidence of serious adverse reactions, and also exhibited adequate pharmacokinetic properties, which warranted its entry into phase II [201].

Ranolazine is a clinically approved drug for arrhythmias and antiangina, which can reduce apoptosis in response to IR damage by delaying the opening of mPTP and improving the structural integrity of complex I [202].

#### 4.1.4. Drugs Targeting Mitochondrial Dynamics

Impaired mitochondrial dynamics affect a wide range of cellular processes such as mitochondrial biogenesis, ROS production, mitochondrial autophagy, and apoptosis, leading to myocardial injury and accelerating the progression of CVDs. Pharmacological interventions that modulate mitochondrial dynamics are therefore considered an effective cardiovascular therapeutic strategy [203].

In CVDs, mitochondrial fission is often abnormally activated, affecting the homeostasis of the heart [204]. For example, mitochondrial hyperfission in the cardiac reperfusion stage of IR rats led to long-term myocardial dysfunction, while targeted inhibition of this process restored the integrity and normal function of the myocardial tissue [205]. The Drp1 inhibitor mitochondrial fission inhibitor 1 (Mdivi-1) improved pressure overload-induced HF by acting as a metastable inhibitor of GTPase assembly to inhibit the GTPase function of Drp1 and reducing the transfer of Drp1 to the mitochondria [206]. The small molecule drug dynasore can inhibit Drp1 and dynamins 1 and 2. Dynasore was shown to improve the survival and viability of primary adult mouse cardiomyocytes under oxidative stress and display a lusitropic effect in explanted mouse hearts [207].

Disorders of mitochondrial fusion proteins are closely related to myocardial hypertrophy, myocardial oxidative damage, hypertension, atherosclerosis, and other CVDs, and therefore have garnered attention as a therapeutic target in CVDs [208]. BGP-15 is a hydroxylamine derivative capable of modulating the GTPase activity of OPA1, activating mitochondrial fusion, and stabilizing the cristae membrane [209]. BGP-15 inhibited oxidative stress-induced mitochondrial fracture in WRL-68, C2C12, and A549 cell lines in vitro and in a PAH model in vivo [209]. During HF, βII protein kinase C (βIIPKC) translocates to the OMM to bind to and phosphorylate MFN1. Phosphorylation of MFN1 by βIIPKC located at S86 in the GTPase structural domain in MFN1 correlates with a decrease in MFN1 GTPase activity. The novel small peptide SAMβA inhibits the interaction of MFN1 with βIIPKC (βII protein kinase C), thereby improving mitochondrial and cardiac function in a rat model of HF [210]. In addition, trimetazidine, a clinical antianginal drug, improved mitochondrial dynamics balance by increasing mitochondrial fusion-related proteins MFN1 and OPA2, and also reduced HF caused by stress overload through glucose uptake via the AMPK pathway [211].

Many other regulating molecules of mitochondrial physiology have been tested in non-cardiovascular systems. Hence their utility in CVDs remains to be determined. Even when inspecting the ones already tested in cardiac systems, there is insufficient evidence for their efficacy in human disease. Improving drug specificity and pharmacokinetic profile may yield better candidates for clinical trials. Furthermore, mitochondrial fission and fusion are two balanced processes that are required to maintain a normal mitochondrial network in cardiomyocytes. Therefore, therapeutic strategies targeting mitochondrial dynamics may be limited to the temporary regulation of acute and non-chronic CVDs. This type of therapy requires additional scrutiny to avoid excessive bias toward either side of the balance.

### 4.2. Mitochondrially Targeted Gene Therapy Strategies

In theory, mitochondria-targeted gene therapy can target specific abnormal mitochondrial genes for repair, thus achieving a true “targeted therapy”. Current mitochondrial-targeted gene therapy strategies include gene editing, ectopic expression of mitochondrial proteins, MRT, and mitochondrial transplantation. However, the application of these strategies in mitochondrial gene therapy is still being explored and optimized.

#### 4.2.1. Mitochondrial Genome Editing

Mitochondrial genome editing refers to the germline transmission of mutant mtDNA haplotypes by intracytoplasmic microinjection of mitochondria-targeted nucleases to modify human syngeneic or oocytes at risk of mtDNA disease in order to exclude them. The identification of mtDNA pathological variants has become routine due to the development of new gene sequencing technologies. As mtDNA mutations need to accumulate to a threshold before they can cause mitochondria-associated diseases, it is a good idea to repair or excise mutated mtDNA by gene editing techniques to reduce the proportion of mutated mtDNA in the overall mtDNA. Currently, mtDNA damage can be repaired using nuclease technology [212,213] and CRISPR gene editing technology [214]. However, prevention strategies for mtDNA-mediated mitochondrial disease may be less desirable due to the heterogeneity of the mitochondrial genome.

#### 4.2.2. Ectopic Expression of Mitochondrial Proteins

Ectopic expression of mitochondrial proteins is achieved by combining mtDNA sequences encoding mitochondrial proteins and mitochondrial targeting sequences and integrating them into the genome via a vector to construct a recombinant plasmid.

Defective ATP synthesis has been rescued by transferring *MTATP6*, which encodes ETC complex V, to the nucleus and successfully importing the protein into mitochondria [215]. However, potential factors such as whether the ectopically expressed protein affects mitochondrial metabolic processes or whether it can be imported into the mitochondria by correctly assembling with the mitochondrial targeting peptide should be fully considered when using this technique. Lavie et al. targeted ectopic expression of REEP1 carrying pathological mutations to mitochondria in primary neuronal cultures [216]. The mutated REEP1 protein isolates mitochondria to the perinuclear region of the neuron and therefore impedes mitochondrial transport along the axon. Their study demonstrated that ectopic expression of mitochondrial proteins can rescue the mitochondria of cells in vitro, but further therapeutic applications will have to be tested in vivo [216]. However, at present, the ectopic expression of mitochondrial protein has not been applied to mitochondrial dysfunction in CVDs, and future exploration and research in the cardiovascular field are required.

#### 4.2.3. Mitochondrial Replacement Therapy

The limitations of conventional gene therapy approaches made them fall short of expectations in the treatment of pathogenic mtDNA-deficient diseases. MRT is an in vitro fertilization technique used to prevent the transmission of mitochondrial diseases. MRT is the most effective technique for preventing inherited mtDNA mutation disorders by blocking the transmission of mitochondrial genetic diseases to the offspring through mitochondrial replacement of oocytes containing mutant mitochondria [217]. There are three main different protocols for MRT: spindle transplantation (ST), protoplast transplantation (PNT), and polar body transfer (PBT). The resulting embryos all have genetic information from the mother and father and healthy mitochondria from the donor.

The two most used techniques for performing MRT in current research are ST and PNT. Tachibana et al. (2009) successfully tested ST in macaques [218]. They transferred the spindle–chromatin complex of the recipient oocyte to the donor oocyte depleted of its own genome (oocyte spindle transfer) to fuse into a reconstructed oocyte prior to fertilization. The reconstituted oocytes from the mitochondrial replacement were able to support normal fertilization, embryonic development, and the production of healthy offspring. However, this technique would face additional technical, efficiency, ethical, social, and policy issues if applied to humans, and long-term preclinical studies of these important and vital issues in clinically relevant non-human primate models are necessary. In contrast, however, the maternal ST technique is a selective reproductive technique performed pre-fertilization, similar to prenatal diagnosis and pre-implantation genetic diagnosis [219]. Therefore, the ST technique may be more promising and easier to implement in humans in the future.

#### 4.2.4. Mitochondrial Transplantation

Given the complexity of mitochondrial biological functions, researchers have begun to consider mitochondrial transfer to rescue damaged cells, that is replacing damaged mitochondria with healthy mitochondria from donor cells. Mitochondrial transplantation involves the isolation of active mitochondria from normal tissues, such as non-ischemic areas, and the delivery of functionally normal mitochondria to damaged tissues and organs by direct injection or vascular delivery. This approach can replace damaged mitochondria, thereby restoring normal mitochondrial structure and function for the therapeutic purpose of reducing myocardial tissue damage and improving cardiac function [220,221].

Scientists have developed cytoplasmic hybrid cell lines as a practical model for mitochondrial disease. This method of fusing normal cells with mtDNA-deficient cells to form healthy hybrids compensates for mitochondrial dysfunction in mtDNA-deficient cells [222]. Lee et al. observed in a co-culture model that exogenous mitochondria were preferentially transported to mitochondria-damaged cells and tissues, which has implications for the delivery of therapeutic agents to sites of injury or disease [223]. Shin and co-workers used a pig model to show that mitochondrial transplantation by intracoronary delivery in the heart is safe, has specific distribution to the heart, and leads to a significant increase in coronary blood flow [224]. Pre-ischemic mitochondrial transplantation with a single or continuous intracoronary injection prophylactically protected the myocardium from IR injury, significantly reduced the infarct area, and enhanced overall and regional heart function [220]. The myocardia of New Zealand white rabbits subjected to ischemic shock were injected with mitochondria isolated from their own pectoral muscles. Injected mitochondria were internalized by cardiomyocytes within 2–8 h. This uptake resulted in improved cell viability that manifested as a reduced infarct area and increased ventricular contractility [225]. Intracoronary injection of mitochondria 2 h into reperfusion was found to significantly reduce the size of myocardial infarction and increase local and systemic myocardial function in Yorkshire pigs [226]. Even more strikingly, skeletal muscle mitochondria from children with myocardial ischemia injected into their own ischemic myocardium led to significant improvements in ventricular function, with many parameters returning to normal [227]. Moreover, there were no adverse reactions or complications related to scarring, intramyocardial hematoma, arrhythmia, etc. [227]. This finding marks the first successful application of this technique in human IR injury and suggests mitochondrial transplantation as a promising treatment modality for other CVDs. However, whether mitochondrial transplantation affects intracellular homeostasis, and their efficacy, compatibility, and safety issues require further consideration.

**Table 2 biomedicines-11-01500-t002:** Therapeutic strategies for targeting mitochondria in CVDs.

Drugs/Therapy	Mitochondrial Target	Disease	Therapeutic Mechanism	Ref.
Elamipretide	ETC	BTHS	Increasing mitochondrial oxygen flux, complex I and IV	[168,169,170]
OP2113	ROS/H_2_O_2_	IR injury	Specific blockade of ROS/H_2_O_2_ production	[179,180]
Idebenone	ETC/ROS	FRDA/HCM/AMI	Increasing ATP synthesis; ROS-AMPK-mTOR axis	[181,182,183]
Carvedilol	OXPHOS	DIC	Lower troponin levels	[185,186,187]
Mito-TEMPOL	OXPHOS	DIC	Scavenging oxygen free radicals	[189,190,191,228]
Propofol	OXPHOS	IR injury	Transcriptional activation of mitochondrial protein LRPPRC	[193]
Diazoxide	OXPHOS	CVD	Turn on ATP-sensitive potassium channel (K_ATP_)and reduces ROS and Ca^2+^-induced swelling	[194]
CsA	The mPTP	IR injury	Inhibition of mPTP	[197]
The mPTP	The mPTP	IR injury	Improves CaCl_2_-induced mitochondrial swelling	[198]
TRO40303	The mPTP	AMI	Delayed mPTP opening.	[199,200]
Ranolazine	The mPTP	Arrhythmia	Delayed mPTP opening; improved complex I	[202]
Mdivi-1	Dynamics	HF	Inhibition of DRP1	[206]
Dynasore	Dynamics	IR injury	Improved survival and viability	[207]
BGP-15	Dynamics	CVD	Increasing OPA1	[209]
SAMβA	Dynamics	HF	Inhibits the interaction of MFN1 with βIIPKC	[210]
Trimetazidine	Dynamics	Angina pectoris	Improves mitochondrial structural and functional damage	[211]
*PCSK9-siRNA*	Autophagy	IR injury	Inhibition of autophagy	[126]
β-hydroxybutyric acid	Acetylation/inflammation	HFpEF	β-Hydroxybutyric acid targets mitochondrial hyperacetylation	[130]
MitoTALENs	MtDNA	CVD	Targeting mutant loci and suppressing mutant gene replication	[212,213]
CRISPR/Cas9	MtDNA	CVD	Gene editing	[214]

## 5. Discussion and Outlook

In the past, considerable effort has been invested in elucidating mitochondrial dysfunction in CVDs and in developing mitochondria-targeted drugs and technologies. However, many challenges remain to be addressed. The failure of conventional mitochondria-targeted delivery drugs to achieve the desired efficacy in clinical trials is due to the fact that most are only symptomatic treatments instead of pathogenic treatments. Furthermore, the complex and diverse pathogenic mechanisms of CVDs cannot be addressed by a single drug directed at the prevention and treatment of mitochondrial damage.

In contrast, gene therapy in theory remains the ultimate strategy for all-cause treatment. To date, mitochondria-targeted gene therapy trials mainly treated diseases affecting specific tissues, such as eye diseases and neurological-related disorders, with genes delivered to only a very limited number of sites. However, CVDs often involve multiple organ systems and require systemic targeted therapies. This significantly increases the number of therapeutic genes used and the cost of treatment. Furthermore, the risk of off-target effects and abnormal immune responses to systemic gene therapy poses a significant challenge. Novel interventional therapies for mitochondrial dysfunction, such as MRT and mitochondrial transplantation, are now being advanced in experimental and clinical studies. To ensure mitochondrial activity and viability, an optimal method for mitochondrial isolation and storage needs to be established. In addition, further research into mitochondrial delivery methods, dosing regimens, efficiency, and side effects, is still needed.

While many challenges remain to be addressed, we anticipate that with further exploration of mitochondrial function and therapeutic potential, future therapies targeting mitochondrial dysfunction may become a useful option for the treatment of CVDs.

## Data Availability

Not applicable.

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
