# Peer review of "Mitochondrial Dysfunction in Cardiac Diseases and Therapeutic Strategies"

_biomedicines, 2023, doi:10.3390/biomedicines11051500_

Round 1
Reviewer 1 Report
The authors comprehensively summarized how mitochondria contributed to pathogenesis of different heart diseases and extensively discussed current therapeutic strategies that focused on ameliorating mitochondrial damage to improve cardiac function in this review article. There are some suggestions for the improvement:
1. For description regarding the SCRAP complex, the authors did not provide the evidence about the role of the SCRAP complex in maintaining mitochondrial protein turnover. Thus, it is confusing to see that: “indicating a role of the SCRAP complex in maintaining mitochondrial protein turnover necessary for normal heart development” (line 81 – 82)?
2. Under “Mitochondrial Function in Mammalian Cardiac Development”, is there evidence about Mitochondrial Function in the heart during childhood?
3. It is questionable that DRP1-controlled process to remove “damaged mitochondria through mitosis” (line 156). Did the authors mean “mitophagy”?
4. It is confusing for the abbreviation of CHD. It indicates congenital heart disease (line 319) in 3.2, while showing coronary heart disease (line 352) in 3.3. Please consider using different abbreviation for coronary heart disease (may use CAD for coronary artery disease).
5. For table 1, the authors should consider reorganizing mitochondrial component based on alphabetical order as ATP, Autophagy, Complex I, Complex II …
6. Please simplify the content for “Therapeutic Mechanism” in table 2.
7. Please have careful proof-reading. There are many errors/mistakes, such as (but not limited to): 1) Mitochondria produce ATP to drive cardiomyocyte contraction, which provides energy for the heart … (line 26); 2) timely correction of mitochondrial dysfunction and improvement of energy production deficits early in the disease becomes … (line 36); 3) The role of PGC-1α in the developing heart has been under intense scrutiny (line 60); 4) a E13.5 (line 75); 5) and is thus hosts … (line 112); 6) …, that is primarily the concerted actions of … (line 147); 7) Impairments in MQC control mechanisms (line 148); 8) DRP1 is the major known pro-fission protein (line 155); 9) Cardiac diseases phenocopy … (line 181), etc.
Reviewer 2 Report
I really like this work, but I think minor corrections can improve this manuscript:
- Paragraph 2.1 Embryonic stage : I think authors need to amplify the concept related to calcium and Mitofusin 2 role as validated in (Ballard A. JBC 2020)
-Paragraph 2.3 Mitochondria function in adult heart: Authors need to speculate more related to adult heart and MFN1 and MFN2 (Moshi Song Cell Metabolism 2017). Moreover there are a new field related to mitochondria clumps and disease (Franco A. Life 2022)
- Paragraph 3.4 Myocardial Ischemia reperfusion: Authors need to better explain the role of PCK9 in this contest
- Conclusion need to me more clear
Reviewer 3 Report
Dr. Huang et al wrote a nice review manuscript about mitochondrial dysfunction in several disease conditions. Although authors put efforts to write a prolonged review, most of part are descriptive without critical thinking and comments. Lack of author’s own contribution was a major concern in this review. There were also many misconceptions in this review.
For example,
Line 118, “The mPTP is a complex of proteins found between the IMM and OM”. This is not a right statement. MPTP is known to be a non-selective pore located on inner mitochondrial membrane.
Line 169-183, “2.4 Mitochondrial Adaptations with Aging”
“ROS production increases in mitochondria and OXPHOS is impaired-----” How aging leads to mitochondrial damage? How aging leads to increased ROS generation?
Most of recent studies about aging and mitochondrial function are missing here.
Round 2
Reviewer 2 Report
No more comments for authors.Author Response
Please see the attachment.

Reviewer 3 Report
This is a revised manuscript. The quality of the manuscript has been improved, but the reviewer still has some minor concerns.
Line 65-67 “Targeted inactivation of Tfam in embryonic cardiomyocytes was lethal, the direct result of elevated ROS products, DNA damage, and severely 67 suppressed cardiomyocyte proliferation [15].”
Something is missing in this sentence.
Line 71-72 “MFN2 was found to be enriched at contact sites between the endoplasmic reticulum and mitochondria”
Mitochondria and ER are connected through MAM (mitochondrial associated membrane). Contact sites usually refer to combination site of outer and inner mitochondrial membrane.
Line 83 “mitochondrial respiratory chain (MRC), ----”
In line 31, authors already defined “electron transport chain (ETC) ---“. It is better to use ETC rather than MRC to keep the consistency
Line 160, Please replace “MRC” with “ETC”
Line 232-233 “With age, mtDNA mutations gradually accumulate, and many fragments are lost----.”
Could authors give more specified information here? As authors already pointed out that mtDNA encoded 13 subunits of ETC. Did aging leads to alteration in mt-DNA encoded subunits?
Line 298-299, “genetic and functional defects of MRC complex I, particularly the NDUFA7 gene. The expression of Ndufa7 ---”
Please only use one abbreviation either “NDUFA7” or “Ndufa7”, not both.
3.1.1. Hypertrophic Cardiomyopathy
In this paragraph, authors emphasized the importance of mtDNA alteration in cardiomyopathy. Interestingly, NDUFA7 is a nuclear gene encoded subunit. Authors should give more information about the effect of transverse aortic constriction on the alteration of nuclear genes here.
Line 596-597 “For example, vasodilators (such as organic nitrates, molsidomine, etc.) can stimulate the production of ROS,----”
Please give brief explanation how vasodilators increase ROS generation.
4.1.1. Drugs Targeting Mitochondrial Complexes
In this section, please including more mitochondrial targeted treatments including reversible inhibition of complex I using amobarbital or high dose of metformin.
